# Understanding the Phage–Host Interaction Mechanism toward Improving the Efficacy of Current Antibiotics in *Mycobacterium abscessus*

**DOI:** 10.3390/biomedicines11051379

**Published:** 2023-05-06

**Authors:** Mylene Gorzynski, Katalla De Ville, Tiana Week, Tiana Jaramillo, Lia Danelishvili

**Affiliations:** 1Department of Biomedical Sciences, Carlson College of Veterinary Medicine, Oregon State University, Corvallis, OR 97331, USA; 2Department of Microbiology, College of Science, Oregon State University, Corvallis, OR 97331, USA; 3Department of Biochemistry & Molecular Biology, College of Science, Oregon State University, Corvallis, OR 97331, USA; 4Department of Bioengineering, College of Engineering, Oregon State University, Corvallis, OR 97331, USA; 5Department of Animal Sciences, College of Agricultural Sciences, Oregon State University, Corvallis, OR 97331, USA

**Keywords:** *Mycobacterium abscessus*, intrinsic resistance, MmpL drug efflux pumps, phage therapy, phage receptors, antibiotics

## Abstract

Pulmonary infections caused by *Mycobacterium abscessus* (MAB) have been increasing in incidence in recent years, leading to chronic and many times fatal infections due to MAB’s natural resistance to most available antimicrobials. The use of bacteriophages (phages) in clinics is emerging as a novel treatment strategy to save the lives of patients suffering from drug-resistant, chronic, and disseminated infections. The substantial research indicates that phage–antibiotic combination therapy can display synergy and be clinically more effective than phage therapy alone. However, there is limited knowledge in the understanding of the molecular mechanisms in phage–mycobacteria interaction and the synergism of phage–antibiotic combinations. We generated the lytic mycobacteriophage library and studied phage specificity and the host range in MAB clinical isolates and characterized the phage’s ability to lyse the pathogen under various environmental and mammalian host stress conditions. Our results indicate that phage lytic efficiency is altered by environmental conditions, especially in conditions of biofilm and intracellular states of MAB. By utilizing the MAB gene knockout mutants of the MAB_0937c/MmpL10 drug efflux pump and MAB_0939/pks polyketide synthase enzyme, we discovered the surface glycolipid diacyltrehalose/polyacyltrehalose (DAT/PAT) as one of the major primary phage receptors in mycobacteria. We also established a set of phages that alter the MmpL10 multidrug efflux pump function in MAB through an evolutionary trade-off mechanism. The combination of these phages with antibiotics significantly decreases the number of viable bacteria when compared to phage or antibiotic-alone treatments. This study deepens our understanding of phage–mycobacteria interaction mechanisms and identifies therapeutic phages that can lower bacterial fitness by impairing an antibiotic efflux function and attenuating the MAB intrinsic resistance mechanism via targeted therapy.

## 1. Introduction

*Mycobacterium abscessus* (MAB) is the rapidly growing nontuberculous mycobacteria that ubiquitously exist in the environment and has been isolated from drinking water, household showerheads, hospital wastewater, and plumbing in large quantities [1]. MAB is an opportunistic pathogen of clinical importance because it can infect and colonize the bronchial airways of patients with cystic fibrosis (CF), chronic obstructive pulmonary disease (COPD), or bronchiectasis [2] as well as cause disseminated infections in immunocompromised and postsurgical patients [3,4]. Incidents of MAB pulmonary infections have been increasing in recent years [5], causing fatality in 15% of patients [6,7]. While more information is needed on the epidemiology of nontuberculous mycobacteria (NTM), the global prevalence of MAB infections could be explained due to the rise of immunocompromised individuals, increasing population age, and environmental factors that promote a high burden of NTM in nature, increasing the risk of potential exposure [8,9].

MAB belongs to the most difficult-to-treat mycobacterial infection [5,10]. Current treatment regimens for MAB pulmonary infections are prolonged intravenous therapy using a combination of macrolide, aminoglycoside, and *β*-lactam antibiotics, and, even then, with a success rate only in 45% of patients [11]. The natural resistance of MAB to the majority of antimicrobials has limited its therapeutic options [5]. The mechanisms of this intrinsic resistance are multifold and largely associated with a highly impermeable mycobacterial cell wall and the drug influx/efflux systems [5,12,13]. MAB efflux systems have been shown to be major contributing factors to the intrinsic resistance for aminoglycoside, macrolide, and quinolone classes of antibiotics [5,14,15]. Conversely, the use of efflux pump inhibitors has been demonstrated to enhance the susceptibility of bacteria and improve conventional antibiotic therapy [16].

Furthermore, the formation of biofilms is a significant part of MAB pathology, hindering antibiotic efficacy in patients. For example, biofilms are found in the lung cavities of individuals with chronic obstructive pulmonary disease and the intra-alveolar walls [17,18]. The evidence suggests that MAB forms robust biofilms in the cystic fibrosis sputum medium (CFSM) that mimics the lung environment of CF patients [19]. The biofilm condition stimulates an increased expression of metabolic enzymes for oxidative phosphorylation, nitrogen metabolism, and peptidoglycan synthesis [20,21], while also promoting the development of invasive (intracellular) variants of MAB with rough morphology [22]. The environment within biofilms stimulates the metabolic changes in bacteria that significantly impact the effectiveness of antibiotics [18]. Furthermore, in a low oxygen condition within lung granulomas of cystic fibrosis patients, MAB survives by shifting the aerobic to anaerobic metabolic state [18,23], influencing the selection of drug-tolerant/persistent subpopulation of MAB and reducing the efficacy of antibiotics [21,24].

The rapid rise of antibiotic-resistant bacteria has prompted increased interest in using new therapies for the treatment of chronic and refractory infections. Therapeutic bacteriophages (phages) are promising alternatives to antibiotics and have been used during difficult-to-manage cases such as multidrug-resistant and disseminated infections [25,26]. Substantial research has been conducted to demonstrate the efficacy of phages against ESKAPE pathogens in vitro and in vivo [27]. The lytic mycobacteriophages have been shown to effectively kill tuberculosis and NTM organisms in vitro [28], including intracellular *M. tuberculosis* and *M. avium* in macrophages, and C57BL/6 mice when the TM4 lytic phage is delivered via transiently-infected *M. smegmatis* [28,29]. The potential benefits of phage therapy against drug-resistant *M. abscessus* isolates have been demonstrated in the zebrafish model as well [30]. Phages are especially effective when combined with antibiotics [31,32]. For example, the phage interaction with *P. aeruginosa* OprM outer membrane porin of the multidrug efflux system results in a decrease in antibiotic efflux and improved drug efficacy [33]. *Acinetobacter baumannii* phages ΦFG02 and ΦCO01 steer mechanisms that result in bacterial capsule loss and subsequent susceptibility to selected antibiotics and infection with other phages [34]. Furthermore, individual gene deletions in the AcrABZ-TolC multidrug efflux system drastically reduce infection of *S. enterica* serovar Typhimurium by the flagellum-dependent phage Chi, underscoring the essential role of the multidrug efflux system in the phage adsorption process [35].

The mycobacteriophage discovery project created by our group has led to the creation of an MAB-specific lytic phage library, a valuable tool that can be used in understanding many fundamental questions addressing phage biology, to study molecular mechanisms of phage–mycobacteria and phage-mammalian host interactions and to advance phage therapy by rationally selecting phages for clinical application. This phage collection also allows the creation of several effective formulations of “phage cocktails” that can aid in a potential problem of neutralizing antibodies, which may arise in patients during phage therapy [36].

In this study, we characterized the MAB phage library under various conditions of the host environment and established one of the major receptors that phages utilize during adsorption onto mycobacteria. We also describe an important phage–mycobacteria interaction mechanism that attenuates MAB intrinsic resistance via decreased efflux function that, subsequently, contributes to the improved efficacy of antibiotics. 

## 2. Materials and Methods

### 2.1. Mycobacterial Strains and Culture Conditions

*M. abscessus subsp. abscessus* strain 19977 (hereafter, MAB 19977 or wild-type MAB) with a smooth phenotype was used as a reference strain and acquired from the American Type Culture Collection (ATCC, Manassas, VA, USA). MAB drug-susceptible and resistant clinical isolates [13] listed in Table 1 were obtained from the Cystic Fibrosis Research and Development Program at the National Jewish Health hospital in Denver, CO, USA. All strains were stored at −80 °C and cultured on 7H10 Middlebrook agar or in 7H9 Middlebrook broth (Hard Diagnostic, Santa Maria, CA, USA) augmented with 5% oleic acid and albumin dextrose and catalase (OADC, Hardy Diagnostics, Santa Maria, CA, USA) and 0.5% glycerol. 

Bacterial inoculums were prepared in the mycobacteriophage buffer (MP buffer) using MAB of the mid-log phase growth (3–5 days). The MP buffer was made, as previously described [29], with 10 mM Tris pH 7.6, 100 mM NaCl, 10 mM MgSO_4_, and 2 mM CaCl_2_ [37]. Samples were passed through a syringe ten times and adjusted to McFarland Standard 1.0 (approximately 3 × 10^8^ bacteria/mL) or to OD_600_ of 0.5. Bacterial serial dilutions were plated on 7H10 agar for the colony-forming unit (CFU) counts to record the exact concentrations of MAB inoculums for all experiments. 

### 2.2. Creation of MAB Phage Library

Soil and water samples were collected in 50 mL conical tubes from various geographical regions of North and South America, and Asia (USDA permit P526-190513-011). Twenty-five ml of MP buffer was added to soil samples, and water samples were processed directly. Tubes were shaken for 24 h at room temperature and later centrifuged gradually at 3600 rpm for 30 min twice and then at 15,000 rpm for 20 min. Samples were filtered with a 1 µn syringe filter unit first and then with a 0.2 µm filter unit. The purified preparations were spot plated to recheck for any contamination on 7H10 agar plates and then used for the phage isolation process in *Mycobacterium smegmatis* mc^2^ 155 strain with the double-layer agar plating method, as described below. 

### 2.3. Phage Isolation and Propagation

*M. smegmatis* mc^2^ 155 was grown until the mid-log phase (3 days) on 7H10 Middlebrook agar containing OADC supplement and 0.5% glycerol. A bacterial suspension of 3 × 10^8^ CFU/mL was prepared by adjusting turbidity to the McFarland Standard of 1.0. The cleared soil/water preparations (500 µL) were added to bacterial inoculum (100 µL) and incubated at room temperature for 30 min. Sample–bacteria mixtures were added to 4 mL of the top agar consisting of 7H9 Middlebrook broth, Bacto agar, glycerol, and 5% OADC. The top agar was overlaid onto 7H10 Middlebrook agar plates and incubated at 37 °C for up to 4–5 days. If plaque formation was noted, the single plaques were incised from the agar and incubated in MP buffer at 4 °C. After 24 h, phages were re-propagated in *M. smegmatis* for higher titer recovery [37]. The final phage preparations were generated by ultrafiltration in 100 kDa membranes that help to concentrate samples as well as filtrate phage preparations from any possible contaminant bacterial factors and endotoxins (typically <30 kDa). The 100 kDa membrane pore allows for crossflow of any material smaller than phage particles and, because membrane pore size provides an equivalent to a spherical particle with a 3-nm diameter, phages are retained in the cartridge [38]. The phage titers were determined through phage serial dilutions. The phage lysates were stored at −20 °C and 4 °C. 

### 2.4. The Generation of MAB-Specific Lytic Phage Library

Lytic phages against MAB 19977 were identified and confirmed with multiple methods: (1) the spot test—spotting phage lysate on bacterial overlays of top agar for phage plaque formation, (2) the phage liquid culturing—a spot plating of the bacteria–phage mixture on 7H10 agar plates after 24 h exposure of bacteria with phages in 7H9 broth, (3) assessing bacterial viability with the optical density (OD_600_), and (4) with the AlamarBlue fluorometric assay after 5-day exposure of phages with bacteria in 7H9 broth. For the phage spot test, same for the plaque formation method, to visually observe the clear inhibition zones generated by lytic phages, 5 µL of phage samples were stamped on the MAB lawn that has been generated by mixing 100 µL of 3 × 10^8^ CFU/mL MAB inoculum with the top agar that was overlaid on 7H10 Middlebrook agar. For the phage liquid culturing assay, 50 µL of phage samples (≥10^10^ PFU/mL) were incubated with 10 µL of MAB inoculum (3 × 10^5^ CFU/mL) for 24 h in 200 µL of 7H9 broth supplemented with 5% OADC at 37 °C (no agitation). The next day, 5 µL samples were spot plated on 7H10 agar plates to assess bacterial number. The reminder cultures were incubated for up to 5 days in the shaker at 37 °C and bacterial ODs were obtained at the absorbance of OD_600_. In addition, to examine bacterial viability, 10 µL of resazurin (50 μg/mL in MP buffer) was added to each well of the same 96-well plates and incubated until an appropriate color change occurred (~3–5 h). The fluorescence was read at 530/590 nm in a Tecan plate reader. Bacterial growth was compared with a control of non-phage exposure. 

### 2.5. Phage Susceptibility Screen and Efficiency of Plating (EOP)

The phage susceptibility was evaluated with a standard plaque assay. In all assays, phage titers were normalized to 10^10^ PFU/mL, and 3 µL were spotted onto top agar overlays containing 100 µL of 3 × 10^8^ CFU/mL bacteria to observe the plaque formation. The EOP was performed for only 17 phages (out of 131) that were selected based on different criteria. These phage lysates were serially diluted by 10-fold and 3 µL were spotted onto top agar overlays of either *M. smegmatis* mc^2^ 155 or MAB 19977 at the above concentration. Plates were incubated at 37 °C for 3 days for *M. smegmatis* or 5 days for MAB. Once bacterial lawns were visible, plates were photographed for plaque formation analysis. The degree of phage virulence was recorded as a ratio between the titer of the phage at the terminal dilution on MAB divided by the titer of the same phage on *M. smegmatis*. The highly virulent phages were classified as 0.1 < EOP < 1.00, moderately virulent if 0.001 < EOP < 0.09, and low virulence with EOP < 0.009.

In addition, in our pilot study using AlamarBlue fluorometric assay and the phage liquid culturing method, we established the fixed concentrations of bacteria and the MAB phage library that resulted in MAB 19977 full clearance. Therefore, in most experiments described in this study, MAB killing by individual phages in the 7H9 liquid culture was performed with a fixed concentration of bacteria and phages. Briefly, 10 μL of 3 × 10^5^ CFU/mL MAB was exposed with 50 μL of 10^10^ PFU/mL phages and cultured in 200 µL of Middlebrook 7H9 broth supplemented with OADC for 24 h, with no agitation. The next day, 3 µL of each sample was spotted onto Middlebrook 7H10 solid media and, 5 days later, plates were assessed for bacterial growth. 

### 2.6. Phage Host Range in MAB Clinical Isolates 

To determine the host specificity, the generated MAB lytic phage library was screened against seven MAB clinical isolates listed in Table 1. As a first step, we performed a standard plaque assay by spotting phages on the soft agar. An inoculum of the desired bacterial strain was created at 3 × 10^8^ CFU/mL. Two hundred microliters of bacterial inoculum were mixed with 4 mL of the top agar and overlaid on a Middlebrook 7H10 agar plate [37]. After the top agar solidified, 3 μL of phage library (10^10^ PFU/mL) was stamped on the surface using a multichannel pipette. Plates were incubated until clear plaques were visible (4–5 days). In the confirmational assay using the phage liquid culturing, 50 μL of phage lysates (10^10^ PFU/mL) and 10 μL of 3 × 10^5^ CFU/mL bacteria were incubated in 200 µL of 7H9 broth supplemented with 5% OADC for 24 h in a 96-well plate format (Corning). The next day, 3 µL of each well was spot plated onto Middlebrook 7H10 solid media for MAB growth. The wells without phage exposure served as controls. The phage host specificity assay for MAB clinical isolates was performed alongside the reference MAB 19977 strain. 

### 2.7. Phage Activity against MAB in Synthetic Cystic Fibrosis Sputum Medium (SCFM)

To test if MAB-specific phages retain lytic activity in the condition that mimics a nutritional environment of CF patients’ pulmonary mucus, MAB 19977 (3 × 10^8^ CFU/mL) was cultured in the SCFM in the 6-well tissue culture plate and incubated at 37 °C for one week (no agitation). The SCFM was prepared as previously described [39]. On day 7, the formation of bacterial biofilms was confirmed by staining the matrix with 0.1% crystal violet (CV) solution for 10 min at room temperature. Next, plates were rinsed three times with distilled water and left to dry for 10 min. CV-stained biofilms were solubilized with 30% acetic acid for 10 min, and supernatants were moved to new clear 96-well flat-bottomed plates for an optical density measurement at OD_570_ using a plate reader (Epoch Microplate Spectrophotometer, BioTek). Bacteria from duplicate wells were collected by centrifugation at 3600 rpm for 20 min, washed twice in SCFM, and adjusted to 3 × 10^8^ CFU/mL using McFarland Standard 1.0. The MAB inoculum of 200 μL was added to 4 mL of the top agar, overlaid on a Middlebrook 7H10 agar plate, and 3 μL of phage library was stamped on the soft agar surface. Plates were visualized after 4–5 days of incubation at 37 °C and the phage lytic properties were recorded as a formation of clear zones on the loan of bacteria. In addition, MAB inoculum of 3 × 10^8^ CFU/mL was diluted in SCFM to the concentration of 10^5^ CFU/mL, and 10 μL of the bacterial solution was mixed with 10 μL of lytic phage library created in a 96-well plate. The phage–bacterial suspensions in 200 μL of SCFM were incubated for 24 h at 37 °C, and, later, 3 µL samples were plated on Middlebrook 7H10 agar plates to determine MAB growth. Bacterial growth was compared with a non-phage exposure group (negative control) or MAB that was diluted in MP buffer and exposed to phages for 24 h in 7H9 broth media (positive control).

### 2.8. Lytic Activity of Phages against MAB in the Metal Mix Media

The metal mix (MX) media is formulated based on the metal concentrations and pH of the mycobacterial phagosome and has been previously described [40]. Single-cell suspensions of mid-log phase grown MAB 19977 were diluted in the MP buffer or MX media. Phage activity was tested by the phage plaque formation and bacterial spot plating methods, as described for the SCFM experiments where bacterial inoculums were made in the MX media instead of SCFM. The assay was carried out in a 96-well plate format and statically, and the non-phage exposure group or MAB inoculum of MP buffer that was exposed with phages in 7H9 broth for 24 h served as negative and positive controls, respectively.

### 2.9. Phage Activity against MAB Grown under Different Carbon Sources

MAB 19977 was cultured in the minimal media (MM; 7H9 base, no OADC or glycerol) supplemented with either 5% OADC and 10% glycerol, 20 mM glucose, or only 10% glycerol, and was incubated at 37 °C for 5 days under agitation. Bacteria were centrifuged at 3600 rpm for 20 min, washed twice with MM, and adjusted to 3 × 10^8^ CFU/mL. The inoculum of 200 μL was mixed with 4 mL of the top agar and overlaid onto 7H10 Middlebrook plates also supplemented with either 5% OADC and 10% glycerol, 20 mM glucose, or only 10% glycerol. The 7H10 agar plates were then spotted with 3 μL of phage lysates and incubated at 37 °C for 4–5 days. Plaque formation was assessed by recording clear zones on the bacterial loan. In addition, MAB inoculum of 3 × 10^8^ CFU/mL was diluted in MM (supplemented with different carbon sources, as above) to the concentration of 10^5^ CFU/mL, and 10 μL was mixed with 10 μL of MAB lytic phage library one by one in 200 μL of MM supplemented with either 5% OADC and 10% glycerol, 20 mM glucose, or only 10% glycerol. The next day, samples were spot plated on Middlebrook 7H10 agar plates for evaluation of MAB growth. Results were compared to MAB growth that was exposed with phages for 24 h in 7H9 broth supplemented with OADC and 10% glycerol.

### 2.10. Phage Activity in a Detergent-Washed and Trypsin-Shaved MAB

The wild-type MAB was grown until mid-log phase in 7H9 broth in two 30 mL conical tubes and after centrifugation at 3600 rpm for 20 min, pellets were either washed four times with a 10% glycerol and 5% Tween 20 solution or incubated with 125 μg of trypsin for 30 min. While the Tween 20 detergent helps to remove loose lipids from the mycobacterial cell wall surface and it is a reversible process, the short time incubation of trypsin shaves/digests surface-exposed proteins permanently. Phage lytic ability was determined with the phage plaque formation and liquid culturing methods, where detergent-washed and trypsin-shaved MAB were used as inoculums. The spot plating assay was carried out in a 96-well plate format and statically. The non-detergent-treated and non-trypsin-digested groups served as controls.

### 2.11. Evaluation of Antibiotic–Phage Combination Treatment Efficacy against MAB

The concentrations of amikacin (AMK; Sigma-Aldrich, St. Louis, MO, USA), cefoxitin (FOX; Sigma-Aldrich, St. Louis, MO, USA), and ciprofloxacin (CIP; Sigma-Aldrich, St. Louis, MO, USA) at which 90% (IC_90_) of MAB 19977 growth is inhibited are 16 μg/mL, 16 μg/mL, and 1 μg/mL, respectively [13]. To test the antibiotic–phage combination effectiveness in MAB, we first determined the antibiotic concentration and phage titration that inhibited ≤50% of MAB (10^5^ CFU/mL) in 7H9 broth culture. Briefly, AMK, CIP, and FOX were serially diluted in the range of 0.1–16 μg/mL, and 100 μL of 10^6^ CFU/mL MAB 19977 was added to 2 mL 7H9 Middlebrook broth containing 5% OADC and 0.5% glycerol. Alternatively, phages were also diluted in the range of 10^4^–10^10^ PFU/mL with MP buffer. The 100 μL of phage lysate dilutions and 100 μL of 10^6^ CFU/mL MAB 19977 were added to 2 mL 7H9 Middlebrook broth containing 5% OADC and 0.5% glycerol. Tubes without antibiotic or phage addition served as MAB growth control. 

In consecutive experiments, the fixed concentrations of antibiotic (AMK at 1 μg/mL, CIP at 0.1 μg/mL, and FOX at 1 μg/mL) and phage (10^6^ PFU/mL) at which ≤50% of 10^5^ CFU/mL MAB was inhibited in 2 mL of 7H9 broth were used to identify phages with superior activity in MAB 19977 when compared with the antibiotic or phage-alone treatment groups. Therefore, four culture tubes (growth control, antibiotic, phage, and antibiotic + phage treatments) were prepared in the same manner for every phage tested. Culture tubes were incubated in a shaker at 37 °C for 5 days at which point bacterial growth in the control group (no antibiotic/no phage) and differences between control and experimental groups were visually observed. Next, samples were serially diluted and plated on 7H10 agar plates for bacterial CFU quantification. The percentage of bacterial survival was calculated from the MAB growth control group that received no antibiotic or phage treatment. Experiments were performed in three technical replicates and repeated two times. 

### 2.12. Assessing Phage Activity in the Presence of Efflux Pump Inhibitors

The efflux pump inhibitors (EPIs) verapamil (Sigma-Aldrich, St. Louis, MO, USA), carbonyl cyanide 3-chlorophenylhydrazone (CCCP, Sigma-Aldrich, St. Louis, MO, USA), and reserpine (Sigma-Aldrich, St. Louis, MO, USA) were used in this study. The stock solutions of EPIs were prepared in dH_2_O or dimethyl sulfoxide, as appropriate. The working concentrations were previously established [14,15,16], and verapamil and reserpine were used at 50 μg/mL and 40 μg/mL, respectively, and CCCP at 0.4 μg/mL. MAB growth and viability were assessed for each EPI separately as well as in combination by adding 100 μL of 10^6^ CFU/mL MAB 19977 to 7H9 Middlebrook broth supplemented with OADC. Bacteria were incubated at 37 °C shaker and day 5 growth was compared to the MAB control with no EPI exposure. Further confirmation of viability was performed by plating bacterial dilutions on 7H10 agar plates. 

To assess the phage activity in the presence of efflux pump inhibitors, MAB 19977 was grown in 7H9 Middlebrook broth supplemented with OADC and in the presence of all three EPIs for 3 days at 37 °C, shaken. Next, bacteria were centrifuged at 3600 rpm for 20 min, washed twice, and adjusted to 3 × 10^8^ CFU/mL. The inoculum of 200 μL was mixed with 4 mL of the top agar containing EPIs at working concentrations and overlaid onto 7H10 Middlebrook agar plates. The plates were then stamped with 3 μL of phage preparations and incubated at 37 °C for 4–5 days. The phage plaque formations were assessed based on clear zones, which were compared with the “lysis” control of bacteria grown without exposure to EPIs, and no EPIs were added to the top agar media. In the phage liquid culturing assay, MAB 19977 was diluted to 3 × 10^5^ CFU/mL in MP buffer and 10 μL was mixed with 50 μL of phage samples (10^3^–10^8^ titer range) in a 96-well plate. Two hundred µL of 7H9 broth supplemented with 5% OADC and EPIs were added only to the experimental wells, while the control wells did not receive EPI treatment and were incubated for 5 days at 37 °C (no agitation). Next, plates were read at OD_600_, and the bacteria were serially diluted, and spot plated for growth analysis. A non-lytic phage 22 and MP buffer without any phage addition served as controls. 

### 2.13. Degree of Adsorption

The inoculums of MAB 19977 with and without EPIs exposure were prepared and used at a concentration as described in the bacterial spot plating EPIs assay above. The test was initiated by the addition of phage lysates to 2 mL of 7H9 broth at a concentration equivalent to a multiplicity of infection (MOI) of 10 and incubated standing at 37 °C. To standardize for phage stability, phages were also added to broth without bacteria and assessed simultaneously. At a 2 h incubation time-point, cultures were centrifuged for 20 min at 3600 rpm and 4 °C. The supernatants were sterile-filtered using the 0.2 µm syringe filter unit to remove bacteria and the associated phage particles. The concentrations of free/unbound phages were determined in 100 μL supernatants using the titration assay in *M. smegmatis* and used to calculate the phage adsorption rates and percentage from the control phage supernatants (no bacteria added). The experiment was repeated three times.

### 2.14. Susceptibility of MAB_0937c and MAB_0939 Gene Knockout Mutants to Phage Infection

The transposon library of MAB was created using a method previously described [13]. The MAB_0937c and MAB_0939 gene knockout mutants were screened against the MAB-specific lytic phage library with the phage plaque formation and bacterial spot plating methods as detailed for the MAB 19977 reference strain. The suspension of 3 × 10^8^ CFU/mL from MAB_0937c and MAB_0939 gene knockout mutants were made in MP buffer, and 7H10 agar plates supplemented with 5% OADC and 0.5% glycerol were utilized in both assays. 

### 2.15. Complementation of the MAB_0937c Gene Knockout Mutant and Generation of Overexpression Clone

The complemented clone of the MAB_0937c gene was previously constructed using the pMV306 genome integration plasmid [13]. Since the MAB_0939 gene is 11,094 bp in size, we were not able to generate a complemented clone for the pks mutant. We also would like to highlight that a plasmid transformation of a construct of such size in mycobacteria would be very challenging because of its complex and hydrophobic cell wall.

The overexpression clone was generated as follows: the MAB_0937c gene was amplified from the MAB genomic DNA with forward 5′-TTTTTGAATTCCCGATCTCGTGGTGCGAT-3′ and reverse 5′- TTTTTAAGCTTTTATGTCTGCCGCGCGTTGA-3′ primers and cloned either into the EcoRI and HindIII restriction sites of the pMV261:His mycobacterial shuttle plasmid under the control of the Hsp60 constitutive promoter. The resulting pMV261:His + MAB_0937c and pMV261 base vector were transformed into *M. smegmatis* mc^2^ 155 strain competent cells, which were prepared by washing bacterial pellets four times with a chilled wash buffer (10% glycerol and 0.1% Tween 20). The final pellet was resuspended in 1 mL of 10% glycerol. Two hundred µL of competent *M. smegmatis* cells were electroporated with 7–10 µL of plasmid DNA using the Gene Pulser Xcell™ Electroporation Systems and kept for 5 h at 37 °C in 7H9 broth, shaken. Next, transformed cells were plated on 7H10 agar containing 50 μg/mL kanamycin (KM). After 5 days of incubation at 37 °C, bacterial colonies were screened with PCR for the presence of the KM gene using Forward 5′-GTGTTATGAGCCATATTC-3′ and Reverse 5′-TGCCAGTGTTACAACCAA-3′. The PCR program was as follows: 95 °C for 5 min, 35 cycles of 95 °C for 30 s, 57 °C for 30 s, 68 °C for 1 min, and a final extension of 68 °C for 5 min. The positive colonies were grown in 7H9 broth supplemented with KM 50 μg/mL (KM50) and stored at −20 °C and −80 °C.

To analyze MAB_0937c protein expression levels, we performed the Western blot on three kanamycin-resistant and PCR-positive *M. smegmatis* colonies for the MAB_0937c gene. *M. smegmatis* with empty plasmid served as a negative control. Bacterial colonies were grown in 7H9 broth until the mid-log phase, palleted, and mechanically disturbed in 1 mL of PBS using a bead beater. Lysates were precleared twice by centrifugation at 10,000× *g* for 15 min and filtered through 0.2 µm syringe filters to remove bacterial cell debris. Samples were run on 12.5% SDS–PAGE gel, transferred to a nitrocellulose membrane, and blocked with 3% bovine serum albumin (BSA) for 1h. Membranes were probed with the His-tag primary antibody (1:200) for 2 h, washed three times, and incubated with a corresponding IRDye secondary antibody (Li-Cor Biosciences, Lincoln, NE, USA) at a dilution of 1:5000 for 30 min. The protein of interest was detected using an Li-Cor Odyssey Imager. 

### 2.16. Ethidium Bromide (EtBr) Accumulation Assay

The EtBr assay has been previously established and used for measuring efflux function in mycobacteria [41,42]. At first, we performed this assay using the wild-type MAB 19977, MAB_0937c gene knockout mutant and complemented clone, and *M. smegmatis* mc^2^ 155. In brief, bacteria were grown in 7H9 Middlebrook broth with 5% OADC and shaken at 37 °C to an OD_600_ of 0.8. Cells were then centrifuged at 3600 rpm for 20 min and bacterial pellets were washed in the MP buffer. Bacterial inoculums were adjusted to an OD_600_ of 0.4 in MP buffer supplemented with 20 mM glucose and 1 μg/mL EtBr [43]. In addition, EPIs (verapamil, reserpine, and CCCP at 50 μg/mL, 40 μg/mL, and 0.4 μg/mL, respectively) were also added to the MP buffer where appropriate. Two hundred microliters of bacteria were added to each well of a 96-well plate. The control wells contained the MP buffer supplemented with glucose, and EtBr with or without EPIs but no bacteria were included for blanking. The accumulation of EtBr was recorded with an excitation/emission of 530 nm/590 nm using a Tecan fluorometer. Fluorescence readings were acquired in 5 min intervals at 25 °C. The EtBr accumulation assay was conducted in eight technical and three biological replicates.

To assess the efflux function of MAB_0937c protein, we performed the EtBr accumulation assay in the *M. smegmatis* clone overexpressing MAB_0937c gene as described above, and data were plotted against the control *M. smegmatis* clone containing a skeletal pMV261 plasmid. No EPI treatment was applied in this testing.

### 2.17. Doxorubicin (DXR) Efflux Activity

Prior to measuring the efflux activity, the minimal inhibitory concentration (MIC) for doxorubicin hydrochloride (DXR; TCI America, Portland, OR, USA) was determined using the broth serial dilution method, as previously described for antibiotics [13]. In brief, the inoculum of mid-log-phase-grown MAB 19977 was adjusted to 10^6^ CFU/mL in MP buffer and 100 μL was cultured into 2 mL of 7H9 Middlebrook broth containing DXR in the range of 0.5–80 μg/mL. The culture tubes were incubated in a shaker at 37 °C for 5 days. The bacterial MIC was recorded as the lowest concentration of the drug that minimally inhibited the growth of the wild-type MAB. 

To establish the MAB_0937c efflux activity, the MAB_0937c gene overexpression and the pMV261 control clones of *M. smegmatis* were resuspended in MP and adjusted to an OD_600_ of 0.3. Inoculums were then microcentrifuged for 10 min at 15,000× *g* rpm to pellet bacteria and resuspended in 1ml of MP buffer containing 1 μg/mL DXR. Bacteria were incubated at 37 °C for 1 h, followed by washing, microcentrifugation at 15,000 rpm, and suspending bacteria in MP buffer supplemented with 20 mM glucose in the presence or absence of EPIs (50 μg/mL verapamil, 40 μg/mL reserpine, and 0.4 μg/mL CCCP). One hundred microliters of bacteria were transferred to a 96-well plate. The media with all ingredients but no bacteria were included as a blank for normalization. The DXR efflux fluorescence readings were acquired in 5-min intervals over 30 min using the Infinite F200 fluorometer (Tecan Systems Inc., San Jose, CA, USA) at an excitation/emission wavelength of 460 nm/590 nm, respectively. The efflux assay was conducted in eight technical and three biological replicates.

### 2.18. Assessing Phage Effect on MAB_0937c Efflux Rates

To determine if phages can influence the efflux function of MAB_0937c protein, we performed the DXR efflux assay in the MAB_0937c gene overexpression and the pMV261 control clones of *M. smegmatis* in the presence or absence of selected phages. Bacterial cells were preloaded with 1 μg/mL DXR in MP buffer for 1 h at 37 °C and processed as described above. One hundred microliters of preloaded bacteria were transferred into a 96-well plate and mixed with 100 μL of phages at 10^10^ PFU already distributed in a 96-well plate with glucose of 40 mM concentration (20 mM as final). In the confirmational DXR assay, preloaded bacteria with DXR drug were transferred into a 96-well plate and mixed with 100 μL of phages at 10^10^ PFU already distributed in a 96-well plate with glucose and 10% Tween 80 (5% as final). The DXR efflux readings were acquired in 3- or 5-min intervals over a 30 min period on a Tecan fluorometer. 

### 2.19. Statistical Analysis

The statistical analyses were performed using the Student’s t-test. The graphical outputs represent the mean ± standard deviation and were created using GraphPad Prism 9 software. Results from three technical and three biological replicates were analyzed unless otherwise indicated. A *p*-value of 0.05 was considered statistically significant. *p*-values are listed in the figure legends and presented by an asterisk in the graphical outputs.

## 3. Results

### 3.1. Identification of Phage Specificity and Host Range in MAB

To generate the lytic phage library against mycobacteria, we obtained a large set of soil and water specimens (approximately 1000) from various geographic locations in the United States (Colorado, Florida, Hawaii, Minnesota, Nevada, North Carolina, New Mexico, Oregon, Texas, and Virginia), British Columbia, Chile, India, and the Philippines. Samples were collected from the dairy farms of past and/or suspicious incidences of mycobacterial infections. Out of 1000 samples, approximately 300 phages were isolated in the *M. smegmatis* host, and 131 phages were found to be lytic in MAB 19977 (Figure 1). These phages were tested against seven MAB clinical isolates expressing either rough or smooth morphology. Initially, MAB isolates were sequenced for 16S to verify the species and subspecies match to *M. abscessus* subsp. *abscessus*. Next, the MAB lytic phage library was tested with two methods: (1) the phage lysate spotting on the soft agar overlays of MAB and (2) the spot plating of the phage–MAB mixture on 7H10 agar plates after 24h incubation in 7H9 broth at 37 °C, as detailed in the Materials and Methods Section. Both results were recorded on day 5 post-incubation. The phage susceptibility for each MAB isolate was compared to the sensitivity of the MAB 19977 strain. The lytic phages were defined as those that completely lysed the host on the soft agar as well as completely cleared or substantially reduced the bacterial number that was evident in the spot assay when compared to the MAB growth control without phage exposure. The heat map of Figure 1 shows the lytic activity and host specificity range across seven MAB clinical isolates alongside the reference MAB strain 19977. Overall, the majority of phages displayed wide-range activity in MAB with few exceptions. Twenty-four phages were identified as lytic against all MAB strains tested. Seventeen phages exhibited activity in seven, 33 in six, and 23 in five MAB clinical isolates. A range of lytic activity was recorded in four or fewer MAB strains, depending on phage or bacterial isolate (Figure 1). 

### 3.2. Different Environmental Conditions Affect Bacterial Susceptibility to Phages

The substantial research demonstrates that mycobacterial exposure to limited carbon resources or starvation leads to metabolic remodeling and changes to the cell wall composition [44,45]. To evaluate if MAB exposure to different nutrient sources could affect bacterial susceptibility to 27 phages selected from Figure 1, MAB was cultured in the 7H9-base minimal media (MM) enriched with either 5% OADC and 10% glycerol, 20 mM glucose, or 10% glycerol alone. After a 5-day incubation, MAB inoculum was prepared and tested for phage susceptibility, as described in the Materials and Methods. As shown in Figure 2, similar patterns of phage susceptibility were observed for MAB of glucose or glycerol-only media when compared to bacteria grown in the OADC-enriched media, which displayed a decreased efficiency of plating for phages 184, 677, and 818 (Figure 2A,C). In addition, bacterial colony morphology was also evaluated in all conditions (Figure 2B). MAB colonies from the 7H9 broth culture with 5% OADC and 10% glycerol supplement are smooth, circular, and raised, while colony morphology appears umbonate and circular in MM with 10% glycerol, and raised and circular with an outer semi-transparent layer in MM with 20 mM glucose (Figure 2B).

Furthermore, it is well established that the host environmental conditions alter MAB metabolic regulation and stimulate cell wall remodeling associated with increased survival of the pathogen and tolerance mechanisms to antimicrobials [21,46,47]. To characterize if 27 tested phages will retain lytic activity in different environmental conditions that bacteria encounter within the host, we incubated MAB in the synthetic cystic fibrosis sputum medium (SCFM) mimicking the cystic fibrosis patient’s sputum (promotes biofilm formation) [13,19] or the metal mix (MX), which is formulated based on the intracellular trace elements and pH of the mycobacterial phagosome [48]. The phage susceptibility was tested alongside MAB that was tested in 7H9 Middlebrook broth supplemented with 5% OADC and 0.5% glycerol. Interestingly, MAB grown in biofilm and intracellular conditions showed resistance to several phage infections. For the SCFM biofilm condition, only 20 out of 27 phages tested retained lytic activity, but in the metal mix media, only 5 phages could lyse bacteria (Figure 2C). The decreased lytic activity was also observed for some selected phages.

To evaluate if MAB outer surface-exposed lipids or cell wall integrated proteins may play a role in the phage attachment and adsorption process, bacteria were subjected to a detergent washing procedure with 5% Tween 20 and 10% glycerol or to trypsin shaving. The Tween 20 helps to remove loose surface glycolipids from the outer cell surface, while the trypsin can digest the cell wall of exposed proteins without killing bacteria when used for a limited time. MABs that were washed with the Tween 20 and processed for trypsin shaving were used to prepare inoculums and then subjected to phage infection. Bacterial susceptibility to phage infection is shown in Figure 2C. While no resistance was observed for 27 selected phages tested in the Tween 20-washed group, phages 169, 319, and 466 displayed decreased lytic activity (Figure 2C and Figure 3). Because the Tween 20 reversibly removes surface glycolipids, it was likely that it had partially inhibited MAB infection by some phages. However, because the trypsin enzyme irreversibly damages proteins, we observed bacterial full resistance to three phages (70, 560, and 818) and five phages displayed reduced lytic efficiency (Figure 2C and Figure 3).

### 3.3. Some Phages Accelerate MAB Killing When Combined with Antibiotics

To assess if phages can improve antibiotic efficacy, 27 selected phages were tested in combination with antibiotics. While MAB antibiotic-alone groups were treated with subinhibitory concentrations of AMK, CIP, and FOX, and the phage-alone treatment groups received phages at 10^6^ PFU/mL concentration, the phage–antibiotic treatment groups were supplemented with both phage and antibiotics at the above concentrations. Five days later, the control and experimental samples were photographed as well as serially diluted and plated on 7H10 agar plates for bacterial colony-forming unit counts (CFUs) (Figure 4A,B). The percentage of bacterial survival was calculated, as described in the Materials and Methods. While we tested antibiotic effectiveness in combination with 27 phages, Figure 4 only displays results for phages that resulted in a significant reduction in MAB growth in combination treatment, suggesting a synergism between the phage and antibiotic action.

### 3.4. Inhibition of Efflux Function in MAB Promotes Resistance to Some Phages

To examine if the inhibition of bacterial efflux pump function can alter susceptibility to phage infection, MAB 19977 was exposed to three efflux pump inhibitors (EPIs), and then bacterial sensitivity against 27 selected phages was evaluated in 7H9 broth using the bacterial spot plating method. In the EPIs treatment group, most phages retained lytic activity; however, six phages (91,113, 256, 278, 280, and 559) exhibited a significant decrease in lytic activity when compared to the corresponding control group of the non-EPIs group at a corresponding phage titration (Figure 5A,B).

In addition, since bacterial efflux systems are located and function at the membrane and could be associated with the phage adsorption process as secondary receptors [35,49,50], to evaluate if EPIs potentially inhibit the phage adsorption process on MAB, phage lysates were added to 2 mL of 7H9 broth cultures of MAB at a multiplicity of infection (MOI) of 10 and incubated standing for 2 h at 37 °C. The free/unbound phages were recovered by clearing sample supernatants with centrifugation and filtration through a 0.2 µm filter. The number of adsorbed phages was calculated by subtracting the unabsorbed phage concentration from the control phage titers without bacterial exposure. The results demonstrated that the inhibition of MAB efflux function leads to a significant decrease in the adsorption rates of six selected phages (Figure 5C).

### 3.5. The MAB_0937c/Efflux Pump and MAB_0939/pks Gene Knockout Mutants Display Resistance to the Majority of Phages

The MAB_0937c gene-deficient mutant has been previously studied and shown to have increased sensitivity to antibiotics in vitro and tissue culture [13]. Because the MAB_0937c gene has a 58% protein identity to *M. tuberculosis* MmpL10 (mycobacterial membrane protein large 10 same as Rv1183), we analyzed the genetic region of MAB in more depth. Figure 6A shows the MAB gene organization and comparisons to *M. smegmatis* and *M. tuberculosis* regions. The MycoMar transposon insertion site in MAB_0937c and MAB_3909 genes is presented in Figure 6B. We established that the MAB_0937c gene is a homolog to the *MmpL10* gene of *M. tuberculosis*. The MmpL10 efflux pump is a part of the glycolipid biosynthesis pathway that exports the cell wall glycolipids—2,3-diacyl trehalose (DAT) and polyacyltrehalose (PAT) on the surface of *M. tuberculosis* and *M. smegmatis* [51,52]. While MAB_0937c is involved in the export of DAT and PAT, the MAB_0939 gene encodes the polyketide (Pks3/4) ortholog gene involved in the elongation of mycolipenic acids needed to synthesize DAT within the cytosol. DAT and PAT are essential structural components of the mycobacterial outer cell wall [53]. The schematic model of the biosynthetic pathway of diacyltrehalose/polyacyltrehalose in mycobacteria is shown in Figure 6C.

### 3.6. Phage Sensitivity in MAB_0937c and MAB_0939 Mutants

To evaluate the susceptibility of MAB_0937c/MmpL10 and MAB_0939/pks gene knockout mutants to phage infection, first, we screened 27 selected phages that we characterized in this study. However, because we observed increased resistance of MAB_0937c and MAB_0939 mutants to many phages, these results prompted us to screen the entire MAB phage library against both mutants (Figure 7). For the most part, MAB_0939 (Pks) mutant showed a similar resistance pattern as the MAB_0937c mutant (Figure 7A). Both mutants were resistant to 47 of the same phages out of 131 tested, suggesting that mutants lacking DAT/PAT on the surface were resistant to phage infection. These results demonstrate that DAT/PAT glycolipids are one of the major components that are involved in the initial binding and phage adsorption mechanism on the mycobacterial surface. A set of phages were still able to lyse both MAB_0937c/MmpL10 and MAB_0939/pks gene knockout mutants but with decreased efficiency. These results were recorded with both the spot test and phage liquid culturing methods (Figure 7B).

The complemented clone of the MAB_0937c mutant was also tested for phage susceptibility alongside a mutant (Figure 7B) and was found to display a similar efficiency of plating as reported for the parental MAB 19977 strain (Figure 1), with a few exceptions. This can be explained by the fact that the pMV306 integrated plasmid that was used to generate the complemented MAB_0937c clone does not replace the defective gene, it integrates at the *attB* of mycobacteriophage L5 attachment site in the bacterial genome [55] and it likely alters MAB susceptibility to some phages. Please note that because the MAB_0939 gene is 11,094 bp in size, we were not able to generate a complemented clone for the pks-deficient mutant.

### 3.7. Measuring the Efflux Activity of MAB_0937c Protein

The EtBr accumulation and DXR efflux assays were carried out in the wild-type MAB, MAB_0937c knockout mutant, and complemented clone. The results were compared to *M. smegmatis* efflux rates in the presence or absence of EPIs (Figure 8A,B). Both EtBr accumulation and DXR efflux assays are well established for measuring the efflux pump activity in bacteria including *M. smegmatis* [56,57]. As shown in Figure 8, substantial changes were observed in the efflux rates of the wild-type MAB or *M. smegmatis* when EPIs and no EPI treatment groups are compared for each organism, and greater efflux rates are observed in MAB than in *M. smegmatis* (Figure 8A,B). However, we did not observe differences in efflux rates between the MAB_0937c gene knockout mutant and complemented clone, most likely because of the compensatory mechanism in MAB via other MmpL proteins.

Since the MAB genome contains 31 copies of *MmpL* (mycobacterial membrane protein large) efflux pump genes in comparison to *M. tuberculosis* and *M. smegmatis* that carry 14 *MmpL* genes, we constructed the MAB_0937c gene overexpression clone in *M. smegmatis*. Three kanamycin-resistant and MAB_0937c gene-positive *M. smegmatis* colonies (validated with the PCR) were subjected to Western blot analysis to determine the MAB_0937c protein expression levels (Figure 8C). To carry out the experiment that could explicitly measure the efflux activity of MAB_0937c protein, we utilized *M. smegmatis* MAB_0937c overexpression clone #3 and measured the *MmpL10* gene efflux function using both the EtBr accumulation and DOX efflux assays (Figure 8D,E). Our results demonstrate that the overexpression of the MAB_0937c gene in *M. smegmatis* induced less accumulation of EtBr or greater efflux of DXR when compared with the *M. smegmatis* clone of pMV261 skeletal plasmid (Figure 8D,E) and suggest the MAB_0937c gene role in MAB efflux.

### 3.8. The MAB_0937c Efflux Function Is Influenced by Some Phages

To evaluate if phages could potentially interfere with and reduce the efflux activity of MAB_0937c, the *M. smegmatis* overexpression clone of the MAB_0937c gene was incubated with selected phages and the efflux activity was measured during the initial 30-min exposure. The efflux activity of the phage-exposed experimental group was compared with the control of the same *M. smegmatis* clone but with no phage exposure (Figure 9). Phages were selected with diverse criteria: either (1) they displayed resistance or reduced efficiency of plating in MAB_0937c/MmpL10 gene knockout mutant and in MAB_0939/pks mutant (phages 70, 380, 457, and 818), and/or (2) diminished lytic activity in MAB after exposing with EPIs (phages 113 and 559) and/or (3) exhibited synergy with antibiotics (phages 70, 82, 559, and 818). The efficiency of plating of selected 17 phages is detailed in Figure 9.

To highlight, we were not able to use the EtBr accumulation assay for this experiment because adding the EtBr to bacteria that were exposed to phages at the same time resulted in the phage DNA labeling outside bacteria, causing inaccurate results. However, rather than measuring the accumulation of the compound, we recorded the efflux rates using the DXR assay. We found two phages (70 and 818) that significantly reduced the efflux rates in the MAB_0937c overexpression clone in comparison to the control group and other phage treatment groups (Figure 10A). We also observed a temporal decrease in efflux activity with 311, 253, and 599 phages at 12 min and later.

Furthermore, to confirm the direct link of 70 and 818 phages adsorption processes with the efflux activity in MAB_0937c, we subjected *M. smegmatis* MAB_0937c overexpression clone to phage treatment in the presence of 5% Tween 80 (Tw80), which fully inhibits the phage adsorption process on bacteria [58]. The phage-exposed groups without Tw80 treatment served as controls alongside no phage with Tw80 and no phage with EPIs control groups of the *M. smegmatis* MAB_0937c overexpression clone (Figure 10B). As shown in Figure 10B, the Tw80 addition to both phage-exposed groups substantially increased DXR drug efflux in MAB_0937c overexpressed bacteria when compared to both phage treatments in the absence of Tw80 and the no phage/EPIs-exposed group. This data, together with the phage susceptibility data of the MmpL10 gene knockout mutant, support the hypothesis that the efflux activity in mycobacteria can be affected by some phages during an initial interaction of phage with the host. This interface potentially increases bacterial susceptibility to antibiotics, conferring improved efficacy of phage–antibiotic combination treatments, as shown for phages 70 and 818 in Figure 4A.

## 4. Discussion

Lytic phages are promising alternatives to antibiotics that can infect and rapidly kill species-specific bacterial hosts without negative effects on mammalian cells [59]. Phage therapy is emerging as a novel strategy to save the lives of patients suffering from drug-resistant, chronic, and disseminated infections [26,60,61,62]. The recent renewed interest in phage therapy has led to a proliferation of new research, venture capital investment, and even clinical trials for phage therapies [25]. The personalized single or “phage cocktails” were developed and used successfully to treat multidrug-resistant infection of *Acinetobacter baumannii* in a patient with necrotizing pancreatitis [61], a chronic infection of *Pseudomonas aeruginosa* in a patient that had undergone aortic arch replacement surgery [62], and *M. abscessus* as well as other NTM chronic infections in cystic fibrosis patients with positive outcomes and no side effects [26]. The use of phages in patients with severe MAB lung infections has been approved by the FDA on a case-by-case basis [26,60], and a recently approved phase 1b/2 trial of phage therapy in cystic fibrosis patients colonized with *Pseudomonas aeruginosa* may demonstrate how phage therapy could become a promising personalized medicine in individuals with prolonged or relapsing infections.

The identification of new phages with specificity and a broader host range and the understanding of how various environmental and mammalian host factors may influence phage activity are important initial steps in the discovery of therapeutic phages that can be successfully used in clinics. The mycobacteriophage discovery project created by our group led us to the generation of the phage library of about 300 lytic phages in the *M. smegmatis* host. To increase the chances of isolation of phages with specificity against pathogenic species of mycobacteria, samples were collected from the dairy farms of past and/or suspicious incidences of mycobacterial infections through established collaborations with local field veterinaries, veterinary technicians, and scientists. Out of 300 phages, 131 phages were found to be lytic in MAB 19977. Because phages are highly selective and kill organisms within a narrow taxonomic range, we evaluated the MAB 19977-specific phage library against seven MAB isolates and established several phages with a wide host range across all tested clinical strains.

The environmental MAB is a successful pathogen that can adapt and survive in various stress conditions through its metabolic remodeling. Experimental evidence found by our group [21] and others [18,23,24] demonstrate that diverse conditions within the host facilitate new phenotypes in bacteria. For successful treatment, antimicrobials need to target the pathogen in conditions such as low oxygen levels and the non-replicating state within granulomas. Furthermore, MAB infections often result in biofilm formation in patient airways, and biofilm structures reduce antibiotic penetration, minimizing the efficacy of treatment [63]. Research shows that MAB undergoes metabolic and cell surface remodeling that alters the cell surface glycopeptidolipid presentation within cystic fibrosis airway conditions [44]. Therefore, when selecting phages for future clinical use and during phage–antibiotic combination treatment, it is essential to assess phage lytic ability in these conditions and utilize phages that retain lytic activity against the MAB of non-replicating and biofilm phenotypes. Phages that may carry polysaccharide depolymerase enzymes will be very useful in degrading biofilm structures as well [64,65]. Our results demonstrated that while some phages lose the ability to infect the host, the majority of tested phages still retain lytic activity in the MAB of the cystic fibrosis sputum media condition, which stimulates biofilm formation and the non-replicating state [19,66]. The ongoing genome sequencing project will establish if these phages may possess enzymes that can also affect the integrity of biofilm structures. In addition, genome analysis will provide valuable information to rule out the presence of pathogenic elements such as virulence, antibiotic resistance, or toxin genes. Phages that display these features will be deemed unsafe for in vivo use.

MAB is an intracellular pathogen that can establish its survival niche and persist in innate immune cells, such as macrophages, through many pathogenicity mechanisms, including the inhibition of phagosome-lysosome fusion and the acidification of phagocytic vacuoles. Within the phagosome environment, bacteria shift to a more anaerobic state [1,67]. Although phages cannot penetrate macrophages and the use of phages may not be an efficient option for the direct targeting of intracellular MAB, our previous research has demonstrated a new strategy to deliver lytic phages within phagosome vacuoles via the non-virulent *M. smegmatis* vehicle, allowing to lyse virulent *M. tuberculosis* and *M. avium* in macrophages [28,29]. Furthermore, during the phage internalization process, phages are taken up by phagocytic cells via the endocytosis process and routed to the cytosol where they are found in vacuoles formed from the plasma membrane [68,69,70]. There is a possibility that some phages similar to mammalian cell viruses can escape the endosome degradation pathway and, thus, infect and lyse bacteria in the intracellular environment. The characterization of phage activity in a phagosome environment is still valuable for both possibilities. Interestingly, testing phage activity in the metal mix condition of the phagosome environment [71,72] shows that the majority of phages lose the ability to infect the host bacteria. Only five phages retain lytic activity, and they overlap with phages found to be active in the MAB of the SCFM condition. This fact suggests that selected phages rely on bacterial components for infections that are still present in the biofilm and intracellular state.

There is very little knowledge available on molecular mechanisms of phage interaction with mycobacteria due to their complex outer cell wall structure [73,74]. The majority of studies have demonstrated that phages utilize lipid and carbohydrate moieties such as peptidoglycan or teichoic acid for adsorption on Gram-positive bacterial surfaces [50]. However, the existing evidence highlights that phages can bind to and interact not only with lipids but with diverse cell surface proteins, such as porins [75,76,77], type VII secretion system components [78,79], and the cell division protein in mycobacteria [77]. For example, phage γ of *Bacillus anthracis* uses the protein receptor GamR, and phage SPP1 of *Bacillus subtilis* the protein receptor YueB to adsorb on the host surface [80]. Phages can also utilize the host transport machinery for absorption into the host [81]. *E. coli* phage T6 binds to the outer membrane protein Tsx, which is a nucleoside-specific channel forming porin [80]. To assess if MAB outer surface-exposed lipids or cell wall proteins may play a role in the phage attachment and adsorption process, we subjected bacteria to a 5% Tween 20 detergent washing step to remove loose surface glycolipids from the cell outer surface or used trypsin digestion step to damage the cell wall exposed proteins. The results demonstrated that the majority of phages retain lytic activity after the removal of free surface lipids from bacteria, except for a few that show the decreased efficiency of plating. While we do not observe significant changes in phage susceptibility, it is likely because the removal of surface glycolipids is temporal and is reversed during MAB growth in culture media. However, the trypsin-shaving experiment resulted not only in a reduction in the efficiency of the plating but bacterial full resistance to three phages (70, 560, and 818). This observation suggests that selected phages utilize cell wall proteins as primary receptors for adsorption on the MAB surface.

Substantial research indicates that phage therapy is more effective when used together with antibiotics and phage–antibiotic combinations may display synergy as well [31,32]. The main goal of this research was to identify a group of phages that can synergize antibiotic action in MAB. Rather than solely relying on phage lytic ability, we aimed to take advantage of the phage adsorption mechanism and investigate if phages could influence MAB intrinsic resistance to antibiotics. The fact is that during bacteria–phage interaction as a bacterial defense mechanism, the pathogen can downregulate, mask, mutate, or replace the surface molecules that phages utilize for binding and adsorption. This trade-off mechanism helps the pathogen to resist and escape from phage killing [49]. However, these alterations can be detrimental to bacterial survival because molecules that phages utilize are essential factors that play a role in cell wall maintenance, defense against antibiotics through influx/efflux systems, virulence, or nutrient uptake within the host-deprived environments [82]. Since the surface transport systems, for example, efflux pumps play a critical role in MAB intrinsic resistance to aminoglycoside, macrolide, and quinolone class of antibiotics [5,12,14], finding phages that can interfere with the function of MAB efflux systems will assert an increased efficacy of current antibiotics when used together with antibiotics.

The MmpL efflux pumps play diverse roles in lipid transport, the uptake of nutrients, iron acquisition, or the extrusion of toxic compounds such as antibiotics [83]. The MAB_0937c mutant, previously identified by our group as one of the major intrinsic resistance elements in MAB, was found to significantly influence bacterial susceptibility to antibiotics, and the absence of this gene synergized the killing of intracellular pathogens in human macrophages [13]. These results are consistent with other studies describing the relationship between the inhibition of MmpL efflux function and increased susceptibility to antibiotics while attenuating the survival of intracellular mycobacteria [14,43,56,84]. The bioinformatics analysis of the MAB_0937c gene established 58% identity to the *M. tuberculosis* MmpL10 efflux pump involved in the translocation of acylated trehalose that is produced in the cytoplasm and then transported through MmpL10 into the periplasmic space, where it is converted from DAT to PAT [85]. The disruption of MmpL10 in *M. tuberculosis* prevents the transport of DAT to the cell surface [52]. DAT and PAT are structural components of the cell envelope and have been shown to play a role in intracellular survival and the downregulation of the host immune response. However, its role in virulence is still unclear [53,54]. Furthermore, the MAB_0939 polyketide (Pks3/4) gene is directly involved in the synthesis of DAT. Therefore, we also tested the MAB_0939 gene knockout mutant to establish phage susceptibility patterns and to support our discovery that the set of phages, which no longer could infect both MAB_0937c/MmpL10 and MAB_0939/pks mutants, utilize DAT/PAT as receptors for adsorption onto the mycobacterial cell surface. In addition, a recent study has demonstrated that knocking out the MAB_0939 gene does not impact glycopeptidolipid (GPL) production in MAB; however, it fully abolishes DAT synthesis [51]. This evidence taken together with our results that the MAB_0939/Pks and MAB_0937c/MmpL10 mutants showed resistance to many phages, supports a notion that DAT/PAT is a major receptor in mycobacteria. We also established a group of phages that exhibited reduced lytic activity in both Pks and MmpL10 mutants, which would suggest that these phages may employ alternative mechanisms for adsorption onto the MAB surface.

Phage recognition of the bacterial host is a well-coordinated multistep process. It is known that the primary receptors regulate an early adsorption process (often reversible) of phage onto bacteria. These receptors are mainly exposed on bacterial surfaces (e.g., peptidoglycan or teichoic acids [73], flagella, and adhesins [35,86]). The phage attachment to the primary receptor subsequently initiates adsorption on a secondary receptor, which then leads to changes in the phage tail and priming the DNA release [87]. The transmembrane transport proteins, such as channels, efflux pumps, and pilus gates, often serve as secondary receptors [50]. In addition, some phages can produce depolymerase enzymes to destroy the polysaccharide-based glycan layer to reach secondary receptors [88]. The phage adsorption on a secondary receptor is an irreversible process. The MAB_0937c is likely a secondary receptor for some phages selected in this study.

To assess this hypothesis and to establish a group of phages that could synergize antibiotic action against MAB through decreased efflux activity, we first characterized the MAB_0937c efflux pump using the EtBr accumulation and DXR efflux assays in MAB. Since we did not observe detectable changes in the efflux activity in MAB_0937c gene knockout mutant versus the wild-type MAB (likely due to the compensatory mechanism through other copies of MmpL efflux pumps), we constructed the MAB_0937c overexpression clone in *M. smegmatis* and characterized bacterial efflux rates. The expression of the MAB_0937c gene in *M. smegmatis* resulted in significantly less accumulation of EtBr or greater efflux of DXR when compared to the *M. smegmatis* control of pMV261 empty plasmid, and increased efflux activity was evident due to the MAB_0937c gene expression. Next, to test if the phage adsorption process could alter the efflux function in the *M. smegmatis* MAB_0937c clone, we selected nine phages with diverse criteria. While we observed partial changes in the efflux activity of the *M. smegmatis* MAB_0937c clone during exposure with 311, 253, and 599 phages, the 70 and 818 phages significantly diminished the MAB_0937c efflux activity when compared to a control group without phage exposure. In the following confirmational assays, we re-measured efflux rates of the *M. smegmatis* MAB_0937c overexpression clone in the absence and presence of 5% Tw80, which fully blocks the phage adsorption process. This study found that by inhibiting the phage adsorption process on MAB, a substantial increase in DXR drug efflux was observed in the Tw80-treated *M. smegmatis* MAB_0937c clone when compared to the phage treatment group in the absence of Tw80. Our data support that the efflux activity in mycobacteria can be affected during the initial interface period of phage–mycobacteria interaction.

It has been well documented that phages can utilize the transport machinery as receptors for entry into the host [81]. For example, the phage OMKO1 adsorb onto and utilize the MexAB- and Mex-XY-OprM efflux pumps as receptor binding sites in *P. aeruginosa* and, as a result, renders the pathogen more sensitive to antibiotics. These structures span the inner and outer cell membranes for Gram-negative pathogens. However, the cell wall structure of mycobacteria is a more complex and more efficient barrier than the outer membrane of Gram-negative bacteria, which efficiently limits the access of antibiotics to their cellular targets. How MmpL machinery traverse substrates to the periplasmic space and mycomembrane, and if MmpL systems interact with periplasmic/outer cell wall proteins remain to be determined. Nevertheless, MspA and CpnT have been described as mycobacterial outer cell surface porins [89,90] and may function together with the MmpL efflux systems. Furthermore, we should highlight that some phages possess enzymes such as capsule polysaccharide depolymerases that can strip carbohydrate barriers on bacterial cell surfaces and compromise cell walls [91], making bacterial cell wall transport systems accessible for phage adsorption. In addition, phage endolysins can lyse Gram-positive bacteria from the outside [92,93], and mycobacteriophage endolysins have been demonstrated to exhibit activity against mycobacteria when added to cells [94,95].

This research demonstrates that a subset of phages can alter the efflux function and stimulate an evolutionary trade-off in MAB while increasing the sensitivity to several classes of antibiotics. The testing of antibiotic–phage combination treatments against MAB using 27 phages (characterized by a broad host range activity) established 11 phages, including 70 and 818, that accelerated the pathogen killing at subinhibitory concentrations of antibiotics. The antibiotics (AMK, FOX, and CIP) selected in this study have been shown to upregulate the efflux pump function in bacteria [13] and, likewise, MAB efflux systems play a major role in the intrinsic resistance for aminoglycoside, macrolide, and quinolone classes of antibiotics [5,14,15]. It has been reported that *Mycobacterium tuberculosis* carries a fluoroquinolone efflux pump operon (Rv2686c-Rv2688c) encoding an ABC transporter that confers resistance to ciprofloxacin [96]. Additionally, beta-lactams such as cefoxitin have been shown to increase the expression of efflux pumps [97,98]. Conversely, the use of efflux pump inhibitors can enhance the susceptibility of bacteria to antimicrobials and improve conventional antibiotic therapy [16]. Our results validate the idea that the alterations to efflux activity can increase mycobacterial susceptibility to antibiotics while improving antibiotic efficacy through the phage–antibiotic combination treatment as demonstrated for phages 70 and 818.

## 5. Conclusions and Outlook

*M. abscessus’s* natural resistance to the majority of antibiotics has limited its therapeutic options. While phage therapy shows a huge promise in treating refractory and drug-resistant mycobacterial infections, the use of phages in humans is still very limited due to the lack of understanding of many basic properties of phage biology, and the field requires a more fundamental understanding of the mechanisms at play. In this study, the characterization of MAB lytic phages led us to a discovery of the surface trehalose-based glycolipid DAT/PAT as one of the major phage receptors in mycobacteria. We also established a set of phages that influenced the function of the MmpL10 drug efflux pump and/or in a combination treatment with antibiotics displayed synergism by significantly reducing or completely clearing MAB with subinhibitory concentrations of antibiotics. Understanding the phage-mycobacterial interaction mechanisms has high potential in identifying the therapeutic phages that can lower bacterial fitness by reducing the function of essential factors required for bacterial virulence. Here, we demonstrate that during phage therapy, rather than relying on only phage lytic ability, phages can be rationally selected to impair antibiotic efflux and attenuate MAB intrinsic resistance mechanisms via targeted therapy. In addition, by identifying a subgroup of phages that interact with genetic factors, such as drug efflux pumps, which are common systems across mycobacterial species and are highly expressed during chemotherapy, as well as in varied stress and physiological conditions [99,100], we can establish therapeutic phages that will most likely display a wide-range activity in MAB strains and will make phage therapy broadly applicable in clinics.

## Figures and Tables

**Figure 1 biomedicines-11-01379-f001:**
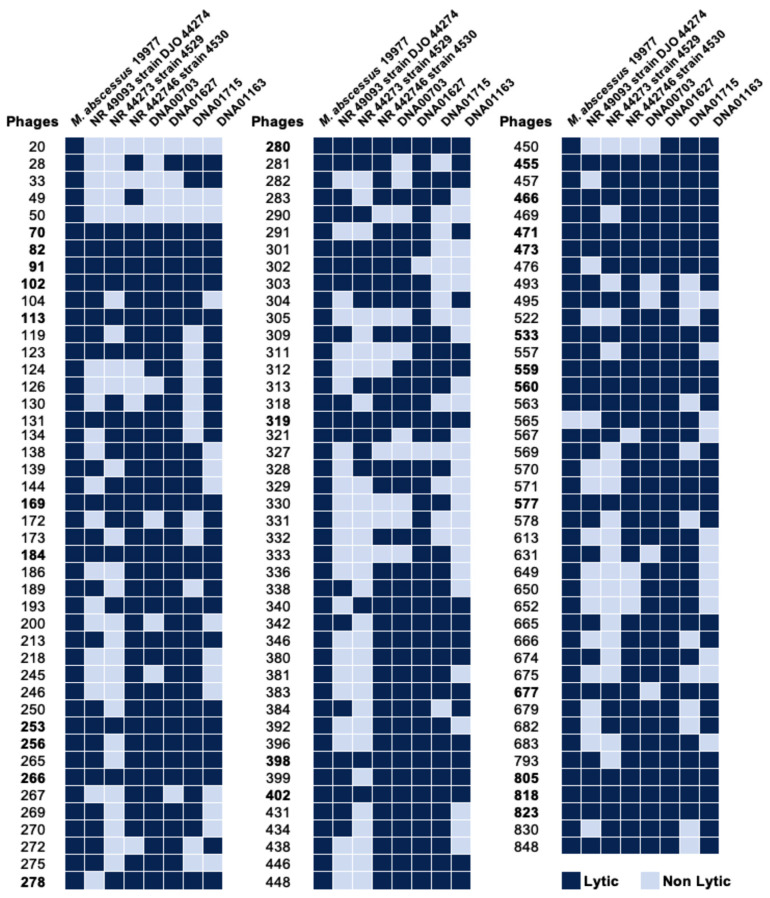
The heatmap of phage lytic activity in MAB clinical isolates. The lytic phage library established in the MAB 19977 strain was also assessed in seven MAB clinical isolates, as described in the Materials and Methods. The data represents a summary of results obtained from the spot test and phage liquid culturing methods. Lytic phages are marked in dark blue and non-lytic phages in light blue. The numbers that are highlighted in bold represent 27 phages that were selected for further analysis.

**Figure 2 biomedicines-11-01379-f002:**
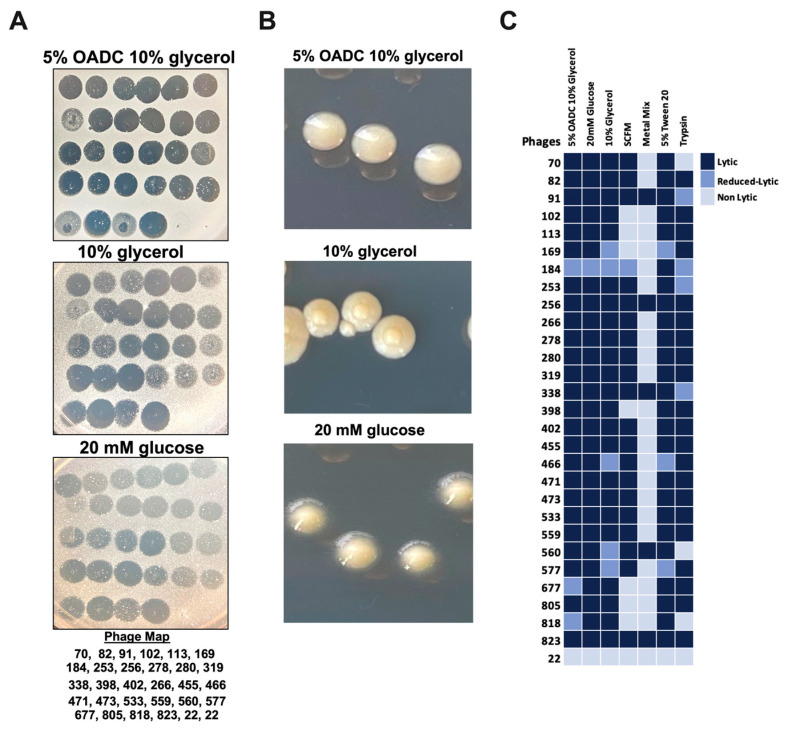
The phage activity in MAB 19977 under different culture and environmental conditions. (**A**) The phage plaque assay was performed by stamping 3 μL of phage preparations on the soft agar that was already mixed with MAB grown in 7H9-base minimal broth supplemented with different carbon sources, as shown in the images. The media content for a specific carbon source was kept constant on 7H10 agar plates as well. Each plaque represents a single phage listed in the phage map. The non-lytic phage 22 was used as a control. (**B**) The colony morphologies are shown for MAB 19977, which was cultured in 7H9 minimal media supplemented with 5% OADC and 0.5% glycerol, 10% glycerol, or 20 mM glucose, and was streaked for colony isolation on 7H10 base agar plates supplemented with the same carbon sources as in 7H9 broth. (**C**) The heatmap represents MAB 19977 susceptibility to phage lysis under various culture, environmental, and treatment conditions, and was analyzed using two methods: the phage plaque assay and liquid culturing. The phage with clear plaques and/or complete bacterial killing activity was considered as a lytic (in dark blue), while reduced lytic activity was recorded with diminished plaque formation and some growth in liquid culture (in medium blue) when compared to the bacterial growth control. The non-lytic phages in light blue displayed no plaque formation and similar bacterial growth in the phage liquid culturing assay as the non-phage-exposed MAB control.

**Figure 3 biomedicines-11-01379-f003:**
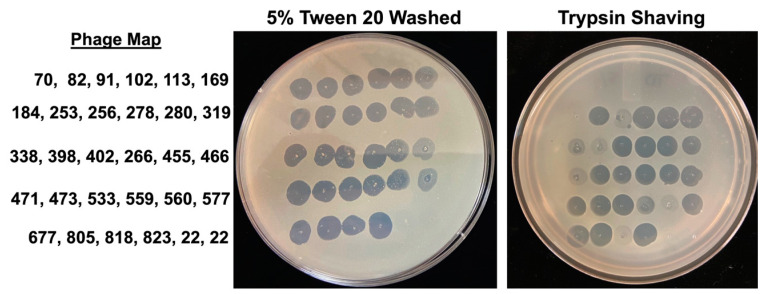
The changes in MAB cell surface composition alter bacterial susceptibility to phage infection. MAB 19977 was washed four times with 5% Tween 20 and 10% glycerol to remove loose outer surface glycolipids or incubated with sequencing grade trypsin enzyme (125 μg/mL) for 30 min at 37 °C to digest cell wall surface proteins. The treated bacteria were then mixed with the top agar and stamped with 27 selected phages. In addition, bacteria–phage exposure was plated on 7H10 agar plates to observe a reduction in bacteria or complete clearance by lytic phages, as described in the Materials and Methods. Phage lytic ability was determined in the same manner as in Figure 2C.

**Figure 4 biomedicines-11-01379-f004:**
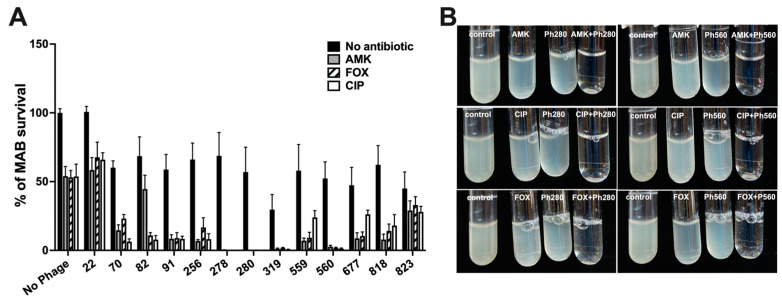
The increased MAB clearance with phage–antibiotic combination treatment. (**A**) The percent of bacterial survival in the antibiotic–phage combination group is shown for phages that accelerated bacterial killing when compared to phage or antibiotic-alone groups. MABs at 10^5^ CFU/mL were challenged with either antibiotic (AMK at 1 μg/mL, CIP at 0.1 μg/mL, or FOX at 1 μg/mL), phages (10^6^ PFU/mL), or antibiotic–phage combinations in 2 mL of 7H9 broth for 5 days. The presence of survival bacteria was calculated by obtaining the CFU counts from serially diluted samples on 7H10 agar plates. Data represent mean ± SD of two biological replicates performed in triplicates. (**B**) MAB growth is shown in culture tubes of antibiotic (AMK, CIP, or FOX), phage (280 or 560), or antibiotic–phage combination challenged groups. The no antibiotic/no phage exposure group serves as the MAB growth control.

**Figure 5 biomedicines-11-01379-f005:**
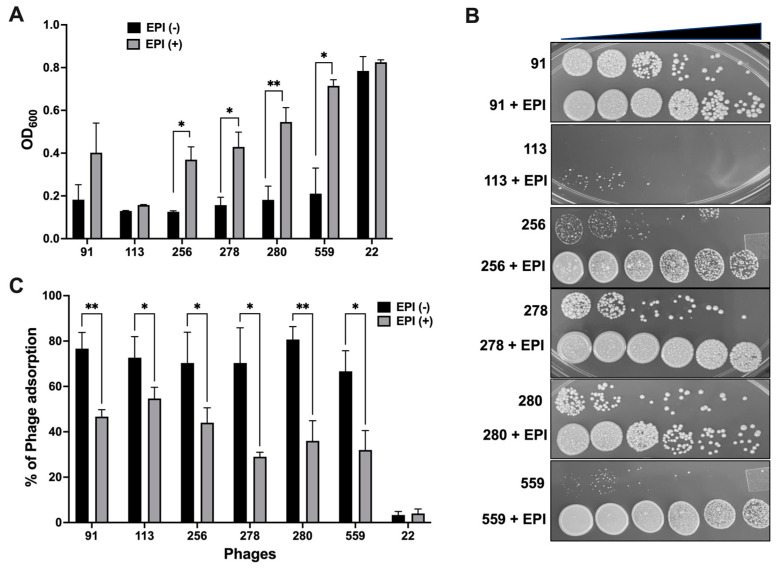
MAB susceptibility to phage infection in the presence of EPIs. (**A**) The OD_600_ was used to measure the growth of MAB 19977 during 5 days of the phage treatment in 7H9 broth in the presence or absence of the efflux pump inhibitors (verapamil, carbonyl cyanide 3-chlorophenylhydrazone, and reserpine). The non-lytic phage 22 served as an additional negative control. The significance was determined using a Student’s t-test (* *p* < 0.05 and ** *p* < 0.01) between EPI and no EPIs treatment groups of three independent experiments performed in triplicate. (**B**) MAB (10^5^) was exposed with phages (in a titer range of 10^3^–10^8^) in 7H9 with and without EPIs supplement and at day 5 spot plated on 7H10 agar to observe changes in bacterial CFUs. (**C**) The phage adsorption rates to MAB were assessed during EPIs exposure. Phages were added to 2 mL of 7H9 broth at an MOI of 10 in the presence or absence of EPIs and free phage titers were quantified in the culture supernatants after 2 h of phage incubation with MAB. The adsorption percentage was calculated from the fraction of unabsorbed phage that was subtracted from viable phages that had been incubated in broth without bacterial cells. All values are the mean ± standard deviation of triplicate assays.

**Figure 6 biomedicines-11-01379-f006:**
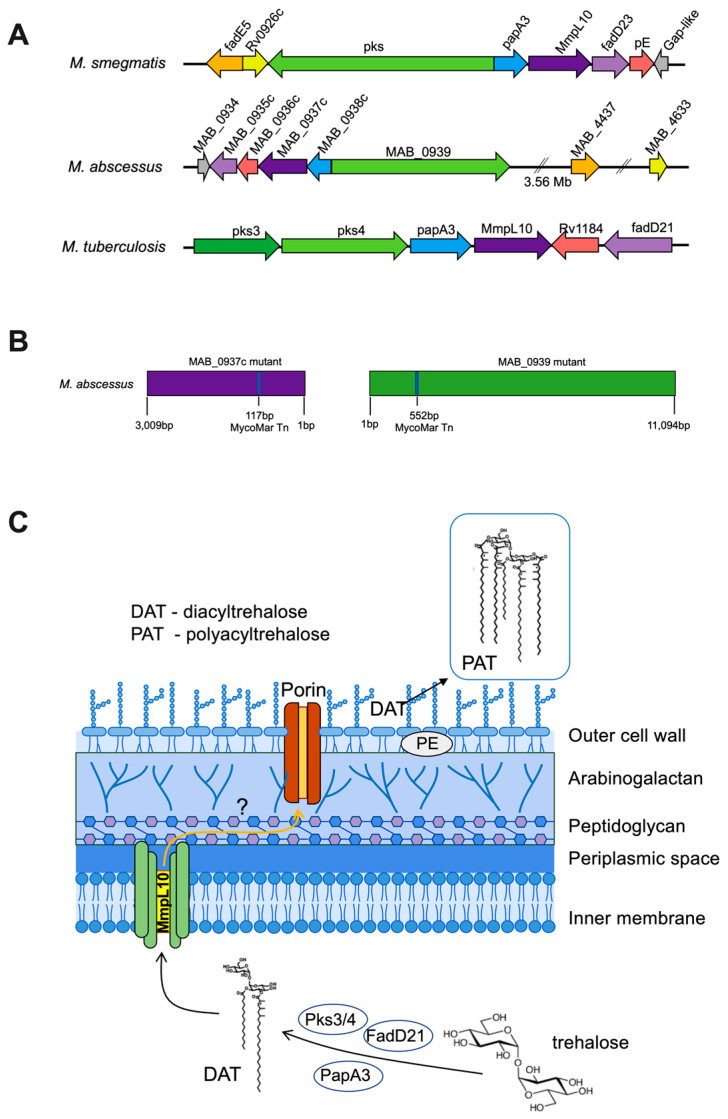
The genomic organization of MAB_0937c and MAB_0939 genes in mycobacteria. (**A**) The genome comparisons for the MmpL10 protein regions of MAB, *M. smegmatis,* and *M. tuberculosis* were obtained by reference articles [51] and the genomic database at the NCBI (http://www.ncbi.nlm.nih.gov/nucleotide). The same color of arrows shows orthologous genes. (**B**) The insertion sites of MycoMar transposon are shown for MAB_0937c and MAB_3909 genes. MAB mutants were sequenced using ligation-mediated PCR (LM-PCR), as previously detailed [13]. (**C**) The schematic representation of the biosynthetic pathway of diacyltrehalose/polyacyltrehalose in mycobacteria. DAT is synthesized in the cytoplasm from the esterification of a straight-chain fatty acid by the PapA3 acyltransferase to the two positions and the three positions of trehalose [54]. The MmpL10 is involved in the transport of DAT across the cell membrane, where it then undergoes transesterification into PAT on the cell surface. The genes involved in this biosynthesis pathway are clustered together in specific genomic regions and are conserved across mycobacteria [51].

**Figure 7 biomedicines-11-01379-f007:**
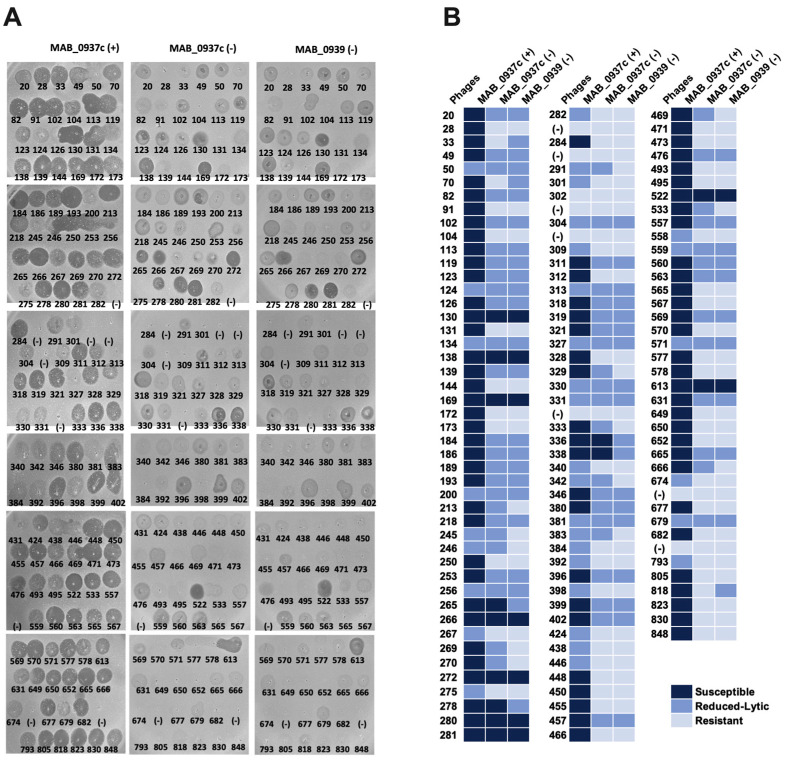
The phage resistance patterns of the MmpL10 efflux pump and pks mutants. (**A**) The MAB 0937c/MmpL10 and MAB_0939/pks mutants were evaluated alongside the complemented MAB_0937c(+) clone in the phage susceptibility assay using the phage plaque formation method. (**B**) The heatmap shows combined results of the phage spot test and phage-bacteria liquid culturing experiments.

**Figure 8 biomedicines-11-01379-f008:**
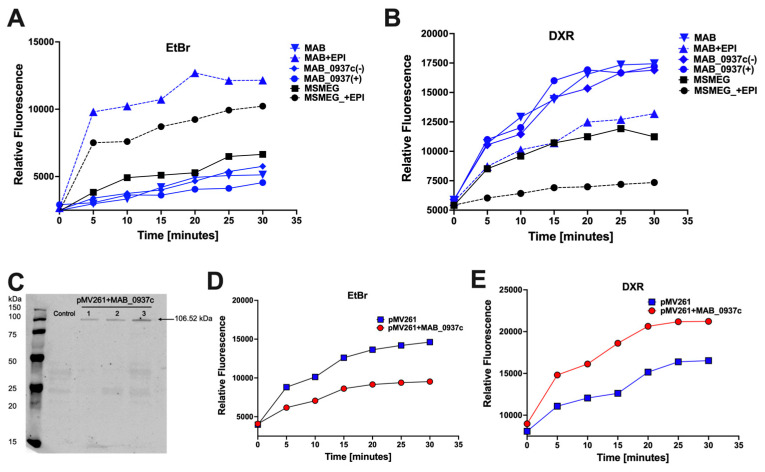
Measuring the MAB_0937c efflux activity. (**A**) The EtBr accumulation assay in *M. smegmatis* (MSMEG) and the wild-type, knockout, and complemented clones of MAB in the presence or absence of EPIs. (**B**) The DXR efflux assay in *M. smegmatis* and the wild-type, knockout, and complemented clones of MAB in the presence or absence of EPIs. For better visualization, MAB groups are marked in blue and MSMEG groups in black. (**C**) The Western bolt analysis of His-tagged MAB_0937c protein in *M. smegmatis* clones. The MAB_0937c protein overexpression clone was constructed using the pMV261:His vector. Three KM50-resistant *M. smegmatis* clones alongside control that contained the empty plasmid were mechanically disturbed using a bead beater. The cleared samples were run on 12.5% SDS–PAGE gel, and proteins on a nitrocellulose membrane were visualized on the Odyssey Imager (Li-Cor) using the His-tag primary antibody and a corresponding IRDye secondary antibody. (**D**) The EtBr accumulation assay was performed for the MAB_0937c overexpression *M. smegmatis* clone (in red) and the control clone carrying empty pMV261 plasmid (in blue). (**E**) The DXR efflux assay was carried out in the MAB_0937c overexpression *M. smegmatis* clone (in red) in comparison with the control clone (in blue). The accumulation/efflux dynamics were measured at 37 °C and in the presence of glucose, and readings were recorded every five-minute interval up to 30 min using the Tecan fluorometer (excitation/emission 530/590 nm for EtBr and 460/590 nm for DXR). The data represent means of eight technical replicates.

**Figure 9 biomedicines-11-01379-f009:**
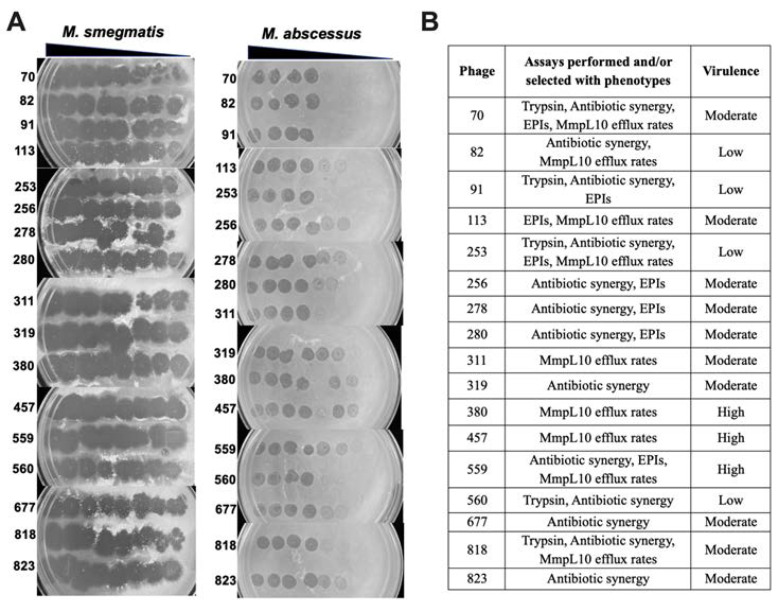
The efficiency of plating for selected phages in MAB. (**A**) The EOP was evaluated by a 10-fold serial dilution of phage lysates and spotting 3 μL of the 10^−1^ to 10^−8^ dilutions onto top agar overlays of *M. smegmatis* mc^2^155 or MAB 19977. (**B**) The degree of phage virulence was recorded for selected 17 phages as a ratio between the titer of the phage at the terminal dilution on MAB divided by the titer of the same phage on *M. smegmatis*. On this basis, phages were classified as highly virulent (0.1 < EOP < 1.00), moderately virulent (0.001 < EOP < 0.09), or low virulence (EOP < 0.009).

**Figure 10 biomedicines-11-01379-f010:**
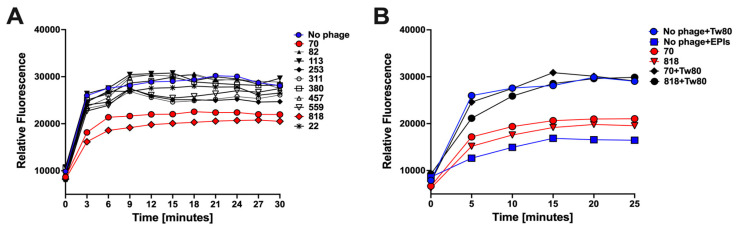
The MAB_0937c efflux activity is affected by phages. (**A**) The DXR drug efflux was measured in the *M. smegmatis* MAB_0937c clone during exposure to lytic phages (in black and red). The non-lytic phage 22 and a group without phage treatment (in blue) were included as controls. (**B**) The DXR efflux activity was evaluated for phages 70 and 818 in the presence (in black) and absence of the Tw80 reagent (in red). The efflux rates of *M. smegmatis* MAB_0937c clone without phage exposure and in the presence or absence of EPIs (in blue) were obtained as positive and negative control data for the DXR efflux assay.

**Table 1 biomedicines-11-01379-t001:** The list of *M. abscessus* clinical isolates used in this study.

Strain	Morphology
*M. abscessus* 19977	smooth
NR 49093 strain DJO 44274	smooth
NR 44273 strain 4529	smooth
NR 442746 strain 4530	rough
DNA00703	smooth
DNA01627	rough
DNA01163	smooth
DNA01715	rough

## Data Availability

Not applicable.

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
