# Peer review of "Understanding the Phage–Host Interaction Mechanism toward Improving the Efficacy of Current Antibiotics in *Mycobacterium abscessus"

_biomedicines, 2023, doi:10.3390/biomedicines11051379_

Round 1
Reviewer 1 Report
The manuscript is devoted to the study of the he phage-host interaction mechanism toward improving the efficacy of current antibiotics in Mycobacterium abscessus. I believe that this manuscript is interesting and should be publishable in this journal; however there are several scientific aspects of this manuscript that I feel the authors must first address.
1. The organization of the introduction of the manuscript should be improved.
2. Please, clarify the choice M. abscessus clinical isolates used in this study
3. Please, clarify the choice of the concentrations of phages using in the work. Why do not use the concentration range of the phages?
4. I suggest authors write a separate 'Conclusions and Outlook' section, to more clearly indicate the value of such studies.
Minor editing of English language required
Author Response
Reviewer 1
The manuscript is devoted to the study of the he phage-host interaction mechanism toward improving the efficacy of current antibiotics in Mycobacterium abscessus. I believe that this manuscript is interesting and should be publishable in this journal; however there are several scientific aspects of this manuscript that I feel the authors must first address.
- The organization of the introduction of the manuscript should be improved.
A: We removed the importance statement and some paragraphs from the introduction. In the introduction, we describe the major treatment challenges for MAB pathogen due to its natural resistance to antibiotics, contribution of efflux pump and biofilms in lowering antibiotic efficacy, and then we introduce the phage therapy as a novel treatment option that is currently used for treating refractory infections. We also added references that highlights some research how phages can synergize antimicrobial treatment.
- Please, clarify the choice abscessus clinical isolates used in this study.
A: M. abscessus clinical isolates were chosen based on their antibiotic susceptibility that is detailed in the reference [13]. We added the “drug-susceptible and -resistant” definition and inserted the reference in the section 2.1. Thank you.
- Please, clarify the choice of the concentrations of phagesusing in the work. Why do not use the concentration range of the phages?
A: The reviewer would agree that performing all assays in concentration range for over 130 phages and across 8 MAB host strains is quite time consuming. The main goal of this study was to narrow down the phage-list and identify and focus on specific phages that can influence MAB fitness while also improving antibiotic treatment outcomes in vitro. Therefore, we performed the EOP assay with and determined the efficiency for only 17 selected phages in a range of titrations and are shown in the Figure 9.
- I suggest authors write a separate 'Conclusions and Outlook' section, to more clearly indicate the value of such studies.
A: As suggested by the reviewer, we added the “Conclusions and Outlook” section. Thank you.
Major changes are marked in red.
Reviewer 2 Report
The article is deal with the understanding the phage-host interaction mechanism toward improving the efficacy of current antibiotics in Mycobacterium abscessus. The topic discussed is very important for the treatment of the pulmonary infections caused by Mycobacterium abscessus.
I would like to make a few comments:
1) The item IMPORTANCE duplicates the information stated in the abstract at the end of the manuscript. It should be deleted.
2) The item 2.2.: 1 μn syringe filter unit …, 0.2 μn filter unit
Check, please, perhaps μm? Indicate, please, the manufacturer of syringe filter units.
3) As far as MAB grown in biofilm and intracellular conditions showed resistance to several phage infection it is necessary to discuss the limitation of the method bacteriophage therapy.
4) There are repetitions in the text, for example in paragraphs 2.11 and 3.3. It is necessary to remove all repeating fragments.
5) It is known that bacteriophages can transmit antibiotic resistance genes. It is necessary to highlight this topic in the discussion and mention the need to study this issue in further research.
6) Conclusion section missing. It is necessary to move part of the material presented in the Discussion section to the Conclusions section.
Author Response
Reviewer 2
The article is deal with the understanding the phage-host interaction mechanism toward improving the efficacy of current antibiotics in Mycobacterium abscessus. The topic discussed is very important for the treatment of the pulmonary infections caused by Mycobacterium abscessus.
I would like to make a few comments:
- The item IMPORTANCE duplicates the information stated in the abstract at the end of the manuscript. It should be deleted.
A: As suggested by the reviewer, we deleted this section.
- The item 2.2.: 1 μn syringe filter unit …, 0.2 μn filter unit. Check, please, perhaps μm? Indicate, please, the manufacturer of syringe filter units.
A: Yes, the reviewer is right. We changed this unite into μm. Thank you.
- As far as MAB grown in biofilm and intracellular conditions showed resistance to several phage infection it is necessary to discuss the limitation of the method bacteriophage therapy.
A: We added this limitation in the discussion section.
- There are repetitions in the text, for example in paragraphs 2.11 and 3.3. It is necessary to remove all repeating fragments.
A: We rewrite the text in 3.3 and removed the experimental details already described in 2.11.
- It is known that bacteriophages can transmit antibiotic resistance genes. It is necessary to highlight this topic in the discussion and mention the need to study this issue in further research.
A: Yes, we agree reviewer that phages can transmit antibiotic resistance and sometimes virulence genes that make bacteria even more pathogenic. This is why is so important to sequence phage genomes and analyze them. We added text and clarified this information in the discussion (page 21).
- Conclusion section missing. It is necessary to move part of the material presented in the Discussion section to the Conclusions section.
A: As suggested by the reviewer, we added the “Conclusions and Outlook” section. Thank you.
Major changes are marked in red.
Reviewer 3 Report
This is a very interesting, well written and comprehensive manuscript. I would advise the authors to rewrite the abstract to include more specific results data rather than just description. Also, manuscript requires some technical editing, i.e. alignment, double space etc. Substantial part of introduction may be placed in discussion section, and some of the results may be included in methods. Due to great length of manuscript it may be an option to present some parts as supplementary data. Results should not include references and such sentences belong in discussion section.
Paragraph one of discussion is missing a reference at the end.
Author Response
Reviewer 3
This is a very interesting, well written and comprehensive manuscript. I would advise the authors to rewrite the abstract to include more specific results data rather than just description. Also, manuscript requires some technical editing, i.e. alignment, double space etc. Substantial part of introduction may be placed in discussion section, and some of the results may be included in methods. Due to great length of manuscript it may be an option to present some parts as supplementary data. Results should not include references and such sentences belong in discussion section.
A: Due to the fact that we studied the library of 131 MAB-specific phages for sensitivity and host range in 8 clinical isolates as well as evaluated their activity in range of environmental conditions, and results show that activity varies depending on the specific phage(s), the abstract will be very descriptive based on naming numbers and statistics. For this reason and because we are also limited in “word numbers”, we prefer to live the abstract as it is. We added a sentence to summarize the phage activity in conditions. We also adjusted formatting, moved some introduction section in the discussion and shortened the results text by removing some methodologies.
Paragraph one of discussion is missing a reference at the end.
A: As suggested by the reviewer, we added the “Conclusions and Outlook” section. Thank you.
Major changes are marked in red.
Round 2
Reviewer 1 Report
The authors considered all my comments. The text has been improved and the manuscript can be recommended for publication in its current form.